# Group Segmentation as a Strategy for Implementing the Intervention Programme in Emotional Education for Infant and Primary Teachers

**DOI:** 10.3390/ijerph192315702

**Published:** 2022-11-25

**Authors:** Miguel Á. Carbonero-Martín, Wendy L. Arteaga-Cedeño, Luis J. Martín-Antón, Paula Molinero-González

**Affiliations:** Grupo de Investigación de Excelencia GIE-179, Departamento de Psicología, Facultad de Educación, Universidad de Valladolid, Paseo Belén N 1, 47011 Valladolid, Spain

**Keywords:** emotional intelligence, intervention programmes, emotional education, sociodemographic profile, teacher work profile

## Abstract

Emotional intelligence is a key social skill for teachers. The teaching profession requires programmes that are geared towards emotional education that will enable teachers to develop emotional intelligence in order to foster their general well-being. The aims of this paper were: firstly, to test the effectiveness of the programme in emotional education implemented through group segmentation based on teachers’ sociodemographic and work profile factors, and secondly, to ascertain whether there were any significant differences in the factors that make up the model of emotional intelligence with regard to the sociodemographic and work variables of teachers in the experimental group in a pre–post analysis study. The design of the research was experimental. The sample was made up of 351 infant and primary education teachers, 190 in the experimental group and 161 in the control group. We used the Trait Meta-Mood Scale-24 together with a questionnaire to define teachers’ sociodemographic and work profiles. Implementing the emotional education programme led to significant differences in the three factors of emotional intelligence (perception, understanding and emotional regulation) depending on the sociodemographic and work profile of the teachers in the experimental group. Applying segmentation allows us to conclude that intervention programmes should be adapted to the sociodemographic and work profile of the participant group. When designing programmes, the method considered should be active, participative, flexible and focused on the teachers’ own experience.

## 1. Introduction

Emotional intelligence has become the best predictor of academic and professional performance [1]. Emotional intelligence is the ability to perceive, use, express and regulate one’s own emotions as well as those of others by processing emotional information [2]. Offering education in emotions is a strategy which the education system aims to implement in order to achieve the general well-being of both students and teachers alike [3,4]. International organisations underpin the key role played by emotional intelligence. The Organisation for Economic Cooperation and Development (OECD) [5] has included emotional skills in the evaluation of teaching staff abilities, since it constitutes the bedrock of students’ psychological–social well-being, and UNESCO [6] has highlighted that emotional intelligence should be embraced within the curriculum as a basic subject. However, emotional education in schools has thus far been approached only scantly and voluntarily in a transversal manner [7,8]. Whilst intervention programmes aimed at students have been implemented at different stages of education, programmes designed for teachers remain few and far between [9,10,11,12].

Given the scarce attention focused on the emotional world of teachers—and due to the insistence on implementing emotional education in the classroom—some authors maintain that teachers should be the primary recipients of emotional education [13,14]. The vast majority of teachers lack both the training and the resources required to develop their emotional intelligence and to apply it in the classroom [3,8]. Teachers’ emotional intelligence impacts their system of beliefs, how they plan their teaching and how they teach in the classroom [13]. Implementing emotional education should form part of teachers’ psycho-educational training so that they can address their own emotions and then provide emotion-related education to their students [7,15]. The few programmes that have been geared towards teachers have shown that educators who display high levels of emotional intelligence are able to respond to the demands of their profession and to apply strategies which counter the emotional distress they experience during their careers [4,16].

Emotional intelligence programmes aimed at teachers have proved to be effective at preventing and/or dealing with aggressive behaviour (physical, verbal, psychological and social), emotional disorders, dissatisfaction with life and professional burnout [4,12,13]. Some authors have explored the influence of emotional education programmes in relation to personal and work variables. Domínguez [17] and Suárez and Martín [18] claim that how emotional intelligence develops differs depending on teachers’ personal and work traits. Muñoz and Bisquerra [19] designed, applied and evaluated a plan in emotional education and found significant differences after the intervention in socioemotional skills, depending on age and gender. Sarrionandia et al. [20] reported that the changes that occurred between males and females after the authors had applied their emotional intelligence programme were similar. Some studies find that women display better levels in factors of emotional intelligence [18,21], whereas other studies find that it is men who do so [22,23]. Analyses carried out in terms of age report that the older the teacher the lower the emotional regulation and the less the interest in perceiving emotions [24,25].

Emotional intelligence has also been linked to teachers’ career profiles. More experienced teachers have been shown to display less emotional understanding [24], while teachers in schools located in rural areas display better levels of emotional perception and regulation [17]. Rojas [26] claims that teachers working in urban areas evidence better levels in the various factors of emotional intelligence compared to their peers who work in rural areas. It has also been highlighted that the level at which teachers teach has no statistically significant effect in terms of socioemotional skills [14,18].

Given the scant number of studies exploring teachers’ emotional intelligence, it is hardly surprising that the findings to emerge thus far have proved contradictory and insufficient as regards ascertaining the true impact of intervention programmes on emotional intelligence, both in the personal and/or work profile as well as in teachers’ overall well-being [3,17,18]. Some authors believe that the emotional intelligence programmes carried out thus far have evidenced a disconnect between real education contexts and research [27,28]. If there is to be a coordinated link between research and the education system, then researchers must view certain factors as essential before, during and after designing and implementing programmes in emotional intelligence [7,29,30].

When devising programmes, physical and psychological aspects must be taken into account as must as participants’ needs and interests as well as their personal and work characteristics [3,17,18,21,31]. This is known as teachers’ sociodemographic and career profiles. Intervention programmes must be taught by professionals who have practical experience in the topics dealt with [8]. Programmes must be evaluated thoroughly and regularly in order to determine their validity and effectiveness [11,29,31,32] Regular programme evaluation helps to pinpoint areas for improvement so as to then enhance the strategies and activities applied during the sessions. Said evaluation also contributes to research aimed at developing and/or strengthening emotional intelligence through intervention programmes. Finally—and as a key element—programmes must be based on a theoretical model in order to determine the aims, content, strategies and activities to be carried out [29,30].

The theoretical models put forward as a basis for intervention programmes differ in the approaches they adopt to study emotional intelligence and socioemotional skills [2,8]. Prominent among the theoretical models are the mixed skills and traits models [33,34]. The theoretical model that had the greatest impact is the Mayer and Salovey ability model [35], and it is considered the model that has been the most researched and to have provided the most contributions to the field of education [36]. The ability model conceives emotional intelligence as the capacity to perceive, utilise, express and regulate one’s own emotions and those of others by processing emotional information [2]. The traits model defined emotional intelligence as the stable traits of personality, behavioural trends and self-perceived abilities [37]. Various authors claim that the Mayer and Salovey ability model is the one best suited for designing, implementing and evaluating emotional education programmes. The ability model has been amply verified, such that it is consistent and can evaluate emotional intelligence through a range of procedures that adopt a solid and robust approach [4,15,36].

This research presents the evaluation of an intervention programme aimed at infant and primary teachers, and which is based on the Mayer and Salovey ability model [35]. The ability model is comprised of four branches. Perception is an individuals’ ability to perceive their own emotions and those of others through gestures and expressiveness. Facilitation is the ability to generate, use and feel emotions, convey feelings and focus attention on important information. Emotional understanding is the ability to recognise emotional information and to grasp emotional meanings. Finally, emotional regulation is the ability to be open to feelings and to promote an understanding towards personal growth. The four factors that make up the skills model are central to the rationale and implementation of the programme analysed in this study. Nevertheless, our research is based on applying a questionnaire developed by Extremera et al. [38] that measures emotional intelligence—TMMS-24 (Spanish version)—grounded on three factors (emotional perception, understanding and regulation). It should be noted that Extremera et al. [39] based their instrument on the original version of the emotional intelligence skills model of Mayer and Salovey [35].

The present study focuses on two main objectives. The first is to test the effectiveness of the programme in emotional education applied through the strategy of group segmentation, carried out taking account of prior analysis of pretest data on emotional intelligence with regard to teachers’ sociodemographic and work profiles. The second aim is to ascertain any significant differences in the factors of emotional intelligence the sociodemographic and work variables of teachers taking part in the training, in line with a pre-post framework analysis.

## 2. Materials and Methods

### 2.1. Description of the Experimental Design for Educational Intervention

We adopted an experimental pre-test and post-test research design with a control group [40]. The intervention applied to the experimental group consisted of an emotional education programme for infant and primary teachers. The dependent variables studied in the control and experimental groups were (overall) emotional intelligence and its dimensions (perception, understanding and emotional regulation). We evaluated the link between teachers’ sociodemographic and work profiles, considering the pre and post-test data of the experimental and control groups. Details of the tools used to collect the information applied in each case are presented later.

### 2.2. Participants

Participants in this study were teaching in the city of El Carmen, in the province of Manabí, Ecuador. The initial number of participants in the study was 417 teachers, of whom 66 (15.8%) voluntarily decided not to continue with their participation as a result of being unable to devote the time required to take part. In this city, there are approximately 1317 teachers at the different levels of non-university education. The final sample was made up of 351 teachers from twenty schools (twelve in the urban area and eight in the rural area), divided into two groups; experimental (54.1%) and control (45.9%). 

The characteristics of the participants’ sociodemographic profile show a clear distribution bias towards women, both in the experimental group (88.9%) and in the control group (87.6%). As regards age, the most represented ranges are 30–39 and 40–49 years of age in both groups. With regard to civil status, many of the participants in the sample were married—both in the experimental group (47.9%) and in the control group (48.4%). In both groups, the most common number of children amongst participating teachers was two or three. As regards the qualifications obtained, a significant percentage of teachers in the experimental group (73.7%) and in the control group (78.3%) held a bachelor’s degree and had not taken further qualifications.

As for the work profile characteristics of participating teachers, most of those in both the experimental group (66.8%) and in the control group (73.3%) were primary school teachers, and most of the participants in the experimental group (84.2%) and in the control group (83.9%) had obtained a fixed post. The majority of participants in the experimental group (70.5%) and in the control group (73.3%) worked in urban areas. In terms of their rank—which reflects their pay grade—most of the teachers in the two groups (experimental group 84.2% and control group 89.4%) were in the lower echelons, since they failed to meet the requirements established for promotion. It is also worthy of note that a large percentage of teachers in both the experimental group (49.5%) and the control group (50.3%) had between 11 and 20 years of work experience. Finally, most of the teachers in the two groups had not held any administrative or managerial posts in education (experimental group 81.1% and control group 79.5%).

### 2.3. Information Gathering Tools

We used the Trait Meta-Mood Scale-24 (TMMS-24) translated and adapted to Spanish by Extremera et al. [39]. The original version is by Salovey et al. [41]. This scale allows us to evaluate the meta-knowledge of emotional states. TMMS-24 evaluates three dimensions with eight items each: perception, understanding and emotional regulation. The TMMS-24 scale is comprised of 24 items, measured on a five-point Likert scale (1 = totally disagree, and 5 = totally agree). The internal reliability achieved by each of the dimensions was: perception and understanding 0.87, and regulation 0.81. Prior studies have evidenced similar data in the internal reliability of the factors—ranging between 0.84 and 0.88 [24,42].

We designed a questionnaire to determine teachers’ sociodemographic profile—such as sex, age, civil status and professional qualifications—and work profile—such as contract situation, the level at which they teach, where the school they work is located, teaching scale category (pay grade), years of work experience and whether or not they have held administrative posts.

### 2.4. Experimental Procedures

The study was approved by the University of Valladolid Ethics Committee. The research complied with the Organic Law Governing Protection of Personal Details in Ecuador—the country in which the programme was applied and in which data was collected. Subsequent to having received approval, we contacted the District Director in El Carmen who oversees and supervises the management of education in order to seek authorisation to enter the schools involved. We also sought the permission of the head teachers in order to hold meetings with teaching staff and to explain to the latter the aims, method and research tools involved and to assure them that any details and information would be treated anonymously and confidentially. We also explained how data would be collected. Participants were requested—through Google Forms—to fill out the questionnaires containing an explanatory text of each tool used, together with a section concerning their informed consent so as to make sure they understood the aims of the research and the voluntary nature of their participation. The pretest was applied to the control group and to the experimental group prior to the start of any intervention, and the post-test was also applied to the two groups once the programme had concluded.

When designing the programme, it was considered important to assess the results obtained in the pretest, as these enabled us to establish group segmentation as an implementation strategy. The analysis allowed us to achieve significant results in the factors of emotional intelligence with regard to teachers’ sociodemographic and work profiles [21].

We met with the heads of the ten schools taking part as the experimental group in order to explain to them how the programme would be applied. The teachers in the experimental group were given 32 h training in emotional intelligence over eight sessions (one per week). Each session was planned to last four hours (three hours of guided work and one hour of self-guided work). The 190 teachers in the experimental group were divided into seven different groups following the segmentations organised for the sessions. It was decided to use group segmentation as a strategy since significant differences in emotional intelligence were found amongst teachers (pretest data) in terms of the sociodemographic and work profile factors, and it was deemed convenient to create heterogeneous groups in which teachers’ emotional intelligence was also strengthened with social reinforcement.

Group segmentation provides positive and significant results in the activities undertaken [43]. We found no studies that have considered a group segmentation strategy to implement programmes in emotional education. Nevertheless, this strategy has yielded excellent results and benefits in other areas of study, such as marketing and business administration, where group segmentation is used as a customer-attention strategy and is seen to provide the attention that is geared directly towards a particular group of people [44]. In order to organise the group segmentation, similar traits displayed by participants are considered (tastes, preferences, interests, difficulties, needs, etc.). Segmentation pursues a group dynamic that is active, meaningful and productive wherein those involved are able to achieve and meet the requirements by examining their individual experiences [45].

Groups were established following heuristic segmentation performed prior to the start of the programme. As can be seen in Figure 1, for the first two sessions, groups were segmented in terms of geographical areas and whether teachers held a permanent position (A). In sessions three and four, participants were segmented depending on the level at which they taught (infant-primary) and their years of teaching experience (B). In sessions five and six, groups were segmented by sex (C). Age and qualifications were taken into account for sessions seven and eight (D). All the segmented groups were structured heterogeneously.

This method of segmenting the intervention groups sought to ensure teacher diversity in terms of their sociodemographic and work characteristics in the various group sessions. In order to define the segmentation variables, we took into account the analysis performed with the pretest data [21] so as to pinpoint which of the teachers’ variables in the sociodemographic and work profiles displayed significant differences vis-à-vis the factors of emotional intelligence.

The programme was carried out online, using Zoom and Teams. Each programme session had a “Teaching Guide” that was provided one day before the session was conducted. The teaching guide was designed to support the teacher training given and to direct the autonomous work. Table 1 provides an example of how session one was designed.

The methodological process that enabled the programme to be conducted was divided into five key moments: first, the initial presentation of the topics and activities to be carried out; second, modelling—setting out the theoretical rationale of the topic; third, guided practice—putting into practice the learning acquired; fourth, feedback—meaningful learning with the whole group through analysis, reflection, debate and consensus, amongst other things; and lastly, the final conclusions derived from each session. The stage of independent autonomous work with emotional intelligence was aimed at being able to transfer and generalise learning. This was performed with the help of the “Teaching Guide”—support material that contained exercises, videos, activities in files, analysis of real situations and of teachers’ own experiences in order to boost emotional intelligence.

The method used to implement the intervention programme in emotional education was practical, experiential, dynamic, participative and interactive [4,8]. Theory and practice were combined, and the work involved both group as well as individual dynamics and reflection, introspection, relaxation, simulation, gamification and real-case analysis, amongst others. Use was made of the “Emotional Diary” for teachers to keep a record of the emotions they experienced at different moments in their life so as to make a personal analysis and evaluate their feelings, thoughts and behaviours. This diary was used as a strategy to develop emotional intelligence in a detailed and personal manner [46]. It is important to point out that teachers had direct access to the activities carried out through links that were included in each planned session in the “Teaching Guide”.

### 2.5. Statistical Analysis of the Data

All of the data to emerge from applying the TMMS-24 were analysed in RStudio v2022.02.3. RStudio Team, RStudio, PBC, Boston, MA, USA [47]. Given the non-normality of the data, we used the Wilcoxon signed rank test, which is a non-parametric test for comparing the mean range of two related samples and determining whether there are any differences between them. In our case, the relation between the samples was given by the experimental pretest and post-test design referred to previously.

## 3. Results

Table 2 shows the pre- and post-correlation between the factors that make up emotional intelligence and which were calculated based on the data derived from applying TMMS-24 to the control and experimental groups, respectively. Broadly speaking, the pre- and post-correlation values amongst the different factors considered are low, both for the control group as well as for the experimental group. 

Our main objective was to evaluate the effect of teachers’ sociodemographic and work profile variables when used as segmentation variables to examine the existence of otherwise significant differences amongst the factors that make up emotional intelligence before (pre) and after (post) training. The results of the evaluation are shown in Table 3 and Table 4. In these tables, a *p*-value equal to or lower than 0.1 should be seen as supporting a statistically significant difference in the corresponding factor of emotional intelligence when the variable in question was used to segment the sample studied.

Table 3 shows the significant differences between sociodemographic profile and emotional intelligence factors. Significant differences emerge in the three factors of emotional intelligence in women. As regards to age, there are significant differences in the ranges from 30 to 50 in emotional perception and understanding, while in those aged over 60 significant differences were found in emotional perception and regulation. Significant differences were obtained in the number of children between those who had no children and those who had one, four and five children in terms of emotional perception, and in those who had two and three children in emotional perception and understanding. With regards to academic qualifications, those who held a bachelor’s degree improved in the three factors, whilst those who had a master’s degree showed significant differences in emotional perception.

Table 4 shows the significant differences between the job profile and the factors of emotional intelligence. Significant differences were obtained in the contract situation variable for teachers working under contract, for those with a provisional appointment in emotional perception, whilst teachers who had a permanent appointment were found to achieve significant differences in the three factors of emotional intelligence. Teachers at both infant and primary levels evidenced improvements in emotional perception and understanding. For their part, teachers working in urban areas achieved significant differences in perception, whilst teachers working in rural areas exhibited significant improvements in the three factors of emotional intelligence. Fewer significant differences in the three factors were seen for teachers at the low end of the pay scale, while teachers in the intermediate pay categories showed significant differences in emotional perception. Teachers with less than ten years of teaching experience, and those with between 10 and 20 years of experience, attained significant differences in perception and understanding, with the latter also achieving significant differences in emotional regulation. Both those who had and those who had not undertaken administrative duties evidenced improvements in emotional perception and understanding, in addition to which those who had not been engaged in such tasks also showed an improvement in emotional regulation.

## 4. Discussion

### 4.1. Effectiveness of the Programme in Emotional Education, Significant Differences in the Emotional Perception, Understanding and Regulation of Infant and Primary Teachers

The results shown in the present study bear out the effectiveness of implementing the emotional education programme, since significant differences were found between pre and post-test in the factors that make up emotional intelligence amongst the teachers who formed the experimental group [35]. This highlights the fact that the infant and primary teachers in the experimental group evidenced significant improvement in terms of developing emotional intelligence. The results from this study prove to be even more valid when we see how teachers in the control group—who did not take part in the programme—experienced no changes, either in overall emotional intelligence or in any of its integrating factors. Our results concur with some other studies which have applied emotional education programmes to an experimental group and which point to significant results when comparing pre- and post-test data [4,20,30,32,38]. They also underpin that the control—who did not participate in the programme—show no significant differences [32,48].

Worthy of note in our results is that not all the participants in the experimental group developed the factors of emotional intelligence equally [12,32]. The factor which displayed the greatest significant difference in terms of its development was emotional perception, followed by emotional understanding and, finally, emotional regulation. These results might well be due to the complexity involved in developing understanding—particularly that of emotional regulation [2]. Factors of emotional intelligence need to be developed in an integrated manner, since each plays a key role in achieving overall teacher well-being [9]. Authors who have applied for a programme in emotional education aimed at teachers have found significant differences in the emotional perception factor [4]. Emotional perception must be developed at a balanced level, since neither too much nor too little attention should be paid to emotions and feelings [14]. Emotional perception is the most important skill to be developed from the ability model, since teachers who display appropriate levels of emotional perception will also exhibit a greater capacity to understand and regulate both their own emotions and those of others [2,41].

### 4.2. Significant Differences in Emotional Perception, Understanding and Regulation with Regard to Infant and Primary Teachers’ Sociodemographic Profile

The second aim pursued by this study focused on determining the significant differences in the factors of emotional intelligence (perception, understanding and emotional regulation) in terms of the sociodemographic and work variables of the teachers who took part. The results show a clear difference in how intervention has an impact with regard to factors of emotional intelligence between men and women. Women obtained significant differences in the three factors (perception, understanding and emotional regulation) whereas men failed to do so in any. These results might be due to the existence of a marked tendency towards female participation in our study (88.9%). In the teaching profession, and in initial teacher training, there is a clear tendency to find women working at a primary level and especially at an infant level [49,50,51]. These data are supported by the statistics published by the Ministry of Education in Ecuador and which state that 71.5% of teachers are female [52]. Similar figures have been reported by the European Office of Statistics [53,54], where 95.2% of infant teachers and 84.7% of primary teachers are women.

As in our study, the levels of emotional intelligence found in other studies with regard to gender show that women obtain better results in the factors of emotional intelligence [21,55]. Other authors, however, report no significant differences in terms of gender [9,56]. Some authors contend that women display better levels in the perception factor, and men in emotional regulation [17,22,23]. Suárez and Martín [18] highlight that women exhibit better levels of understanding and emotional regulation, while Pena et al. [57] maintain that men evidence better levels in understanding and emotional regulation. According to Gallego [58] and Madolell et al. [59], men might display greater emotional control because of family, cultural, and social factors, which have led them to create stereotypes and to respond in a distant emotional and affective manner to their students, children, and loved ones. Reeser and Gottzén [60] claim that men tend to conceal their emotions and feelings for fear of being seen as weak, with this weakness being deemed a threat to their masculinity. It seems clear that women show a greater interest in managing difficult situations by both offering and seeking social support whereas men tend to delegate the responsibility to family members, guidance counsellors, and psychologists [21,61].

We also found significant differences in the factors of emotional intelligence in terms of age. Teachers aged between 30 and 59 recorded significant differences in the factors of emotional perception and emotional understanding, although this was not the case for the factor of emotional regulation. Some authors maintain that there are no statistically significant differences between age and emotional intelligence [10,18,48,62]. Older teachers (30–59 years old) display low levels of emotional regulation, which after the age of 60, tends to increase [63]. This might be because teachers find it difficult to leave personal–family concerns to one side when teaching, and because teachers aged over 60 might feel more free to organise and manage their personal life. This allows us to conclude that during their later adult years teachers combine their personal–family life and their professional life, which to a certain degree affects their emotional state and, above all, their emotional regulation and handling thereof in the classroom.

In our study, we also found that teachers aged over 60 display higher levels of perception and emotional regulation, a result that concurs with some other authors who claim that the older one is the greater the emotional regulation [3,62]. Nolen-Hoeksema and Aldao [25] and García-Domingo [24] contradict these results and maintain that older teachers tend to display less emotional regulation. Márquez and González [64] assert that teachers tend to improve their emotional regulation as they get older due to the physiological factors involved in emotions (a reduction in heart rate when faced with emotions), the subjectivity of emotional experience (fewer negative emotional experiences), the interaction between emotion and cognition (greater relevance of emotional stimuli when processing information) and, finally, subjective emotional control (control of perceived emotions and moderation of the positive affect).

Significant differences were also found in the factors of emotional intelligence with regard to the number of children variable. Specifically, the emotional perception factor displayed significant differences at all levels of the number of children variable. Significant differences were seen to exist in the emotional understanding of teachers who had two or three children, whilst teachers who had no children were able to develop emotional regulation. Some previous studies report that the number of children has no impact on the level of emotional intelligence development [65,66], whilst Arteaga-Cedeño et al. [21] did find a significant and positive relationship between the two variables. Amongst the studies to have explored how family size impacts emotional intelligence, Morand [67], Turculet and Tulbure [68] point to a positive correlation between the two variables. Sánchez-Núñez [69] contends that children have a bearing on the emotional intelligence of their parents. When searching for information, we found few studies that explore how the number of children impacts teachers’ emotional intelligence. We believe that future research should consider such factors when conducting analyses.

The factors of emotional intelligence also exhibited significant differences in terms of professional qualifications. Teachers who hold a master’s degree showed significant differences in emotional perception and emotional regulation, while teachers who hold a bachelor’s degree exhibited significant differences in the three factors of emotional intelligence (emotional perception, emotional understanding and emotional regulation). The fact that teachers who hold a bachelor’s degree and a master’s degree achieve better levels in the factors of emotional intelligence might well be because they are more interested in lifelong learning, given the importance they attach to their profession. In contrast, teachers who hold a degree in teaching (a technical qualification) and those who have no teaching qualifications feel that it is more important to devote time to curriculum alignment. We have failed to find any studies that have explored these factors, such that we highlight the importance of future research looking into them.

### 4.3. Significant Differences in Emotional Perception, Understanding and Regulation with Regards to Work Profile of Infant and Primary Teachers

Significant differences were also found in the three factors of emotional intelligence in terms of teachers’ contract situations. The specific factor that displayed significant differences with all of the variables was emotional perception. Teachers who have been given a permanent position evidenced significant differences in the three factors of emotional intelligence. Nespereira-Campuzano and Vázquez-Campo [70] find that job stability enables workers to display a greater ability to perceive and express their feelings. After having been awarded a fixed post (definitive appointment), teachers exhibit better levels of emotional intelligence [71]. In addition, it should be highlighted that teachers who are under temporary contracts also show significant differences in emotional regulation. We were unable to find any studies that have examined these variables in samples involving teachers that would allow us to compare our findings.

As regards the level of infant and primary education, both groups displayed significant differences in the factors of emotional perception and emotional understanding, results which concur with other studies [21]. There is some degree of controversy surrounding these results, since certain authors find that infant education teachers exhibit better levels in overall emotional intelligence [15], whereas other authors contend that it is primary education teachers who display better levels of emotional intelligence [22,24]. Other studies have found that the higher the level at which teachers teach, the lower the level of emotional regulation they display [63]. It has also been pointed out that, at the infant and primary level, a range of activities and strategies are carried out that directly impact teachers’ emotional intelligence [10]. It could be said that teachers working at the levels considered in our study tend to show a greater interest in lifelong learning [72]. Martínez-Saura [14] states that the level at which teachers teach evidence no statistically significant results in the levels of emotional intelligence.

Thanks to the programme, teachers working in rural areas evidenced significant differences in the three factors of emotional intelligence, whereas those teaching in urban areas only achieved significant differences in the factor of emotional perception. We have not found any studies that have conducted an analysis of these factors among teachers. Nevertheless, we did find some studies that focused on students. Some authors claim that rural areas favour emotional perception and emotional regulation while urban areas favour emotional understanding among students [17]. It has also been pointed out that students in urban areas exhibit higher levels of socioemotional skills compared to students in rural schools [26]. Particular emphasis has been placed on the fact that the social and cultural context impacts the development of emotional intelligence [73].

The results show that the factor of emotional perception obtained significant differences amongst teachers who are at the lowest level of the pay scale; D–E and F–G. The latter also achieved significant differences in the factors of emotional understanding and emotional regulation. The study by De-Haro [74] reported significant differences in pay grade in terms of a person’s job performance and emotional states, while Wu et al. [75] contend that a worker’s salary influences levels of emotional intelligence and job satisfaction. For López and Guiamaro [76], salaries are a source of stimulus and experiences that impact various aspects of the individual, particularly in the socioemotional field.

The ranges considered in teachers’ years of experience pointed to significant differences—specifically amongst teachers with less than ten years of experience—in terms of emotional perception and emotional understanding. Teachers with between 11 and 20 years of experience were seen to display significant differences in the three factors (emotional perception, emotional understanding and emotional regulation). Some studies point to there being no significant differences between years of experience and the factors of emotional intelligence [21,48,66,77]. In the study by García-Domingo [24], teachers who had greater work experience exhibited less attention and emotional regulation. The study by Reza-Amirian and Behshad [78] found that years of teaching experience evidenced significant differences in emotional intelligence. This is because having more work experience and greater emotional intelligence can help teachers who are less experienced. Moreover, emotional intelligence helps to mitigate the negative experiences teachers have had during their years of work experience [79].

Our analysis found significant differences in emotional perception and emotional understanding, both amongst teachers who had and who had not been involved in administrative tasks, with the latter also evidencing significant differences in emotional regulation. Previous studies have shown that teachers who have not undertaken administrative duties display high levels of emotional understanding, unlike teachers who have held some administrative posts [21]. Other studies report that having engaged in administrative tasks influences teachers’ development of overall emotional intelligence [14,18,80]. One might interpret that when teachers hold an administrative post they have no students in their care, which allows them to feel calmer, more relaxed and less concerned about teaching.

The factors we attribute to the results obtained include the theoretical ability model of Mayer and Salovey on which the programme is grounded. This model is both the most widely tested and the most widely used in emotional education programmes [4,14,42,81,82]. Another key factor was the design, implementation and evaluation of the programme (before, during and after it was applied) [11,20,22,31,32,83]. Another important indicator considered when designing the programme was the dynamic, interactive, participative and experiential strategies—considered by some authors to be the most suited for this kind of intervention [8,12,84]. The analysis of pretest data led us to consider the strategy of group segmentation when implementing the programme. The factor we deem to be important is that those who are responsible for training should be specialists in the field in order to ensure that participants stay the course and that they achieve significant learning [85,86,87].

### 4.4. Limitations and Future Lines of Research

Key among the limitations of this study is the use of self-report measures. Another limitation concerns the fact that previous studies evaluating emotional education programmes geared towards teachers have used a small and/or not very representative sample such that it is impossible to ensure applicability, given that it is insufficient for determining whether the results are reliable or not [12]. Finally, it should be highlighted that we failed to find sufficient studies of an experimental nature that would allow us to conduct a more in-depth differential and/or comparative analysis with our findings, thus preventing us from making generalisations concerning the results presented. We suggest that a future inquiry which is focused on training in emotional education should consider implementing existing programmes so as to put forward improvements and thereby achieve a programme that is robust, reliable and easy to apply to teachers, thus ensuring quality rather than seeking quantity [30]. We also believe that future studies should compare and complement the results obtained for teachers with evaluative strategies that consider other sources of information (students and/or parents). In addition, we recommend that a future inquiry should use measures to evaluate the skills learnt during participation in the programmes [38].

## 5. Conclusions

The results obtained from our study, and when compared to those from previous studies [4,20,30,32,38,88], enable us to highlight that implementing emotional education programmes is effective, since it helps to develop emotional intelligence as well as each of the factors that comprise it. This study has shown that prior to designing and putting into practice the emotional education programme, an analysis must be carried out of teachers’ personal traits in order to pinpoint their needs and so gain a clear idea of the level of development of each factor that makes up emotional intelligence. It is also important to gauge participants’ overall level of emotional intelligence, which will help to pinpoint the aspects the programme most needs to focus on, the aims pursued, the content to be applied, which strategies and activities should be chosen and which resources should be used.

Thanks to the analysis of pretest data, we have been able to identify significant differences in the factors of emotional intelligence with regard to sociodemographic and work profiles. These results allowed us to consider a group segmentation strategy when implementing the programme, in which the different sessions involved diverse groups of teachers who had displayed high levels of the factors that constitute emotional intelligence. This enabled us to include teachers who—through their experiences—helped the sessions to enjoy greater participation and for teachers to show greater initiative to comment on situations that affected them when they were working.

As regards what is innovative about our study, particularly worthy of mention is the application of a training programme in emotional education conducted online, which has allowed us to draw on a large number of participants. The online tools used helped to organise the segmented groups and with the correct distribution thereof so as to deliver each of the sessions. It is also worth highlighting that teachers showed interest and that there was active and dynamic participation from all the groups organised. Of the programmes we looked at that contained a similar number of total programme hours, none were carried out online. What we did find was that there were specific activities that had been conducted for certain sessions. We also noted that there had been lectures, seminars, symposiums and activities that had lasted less than ten hours. We believe that it is necessary to implement online tools that prove convenient for the teachers involved so that they can take part in these activities without having to interrupt their personal activities.

We also show that the factor of emotional perception is the one which evidenced the greatest differential development in relation to sociodemographic and work profile variables. Emotional perception is considered to be the most important factor, since developing it allows teachers to recognise, identify and pinpoint the emotions they experience. This aids the development of emotional understanding and, consequently, emotional regulation—factors which prove complex in terms of achieving suitable levels of development. However, thanks to the concepts considered when designing and implementing the programme, we were able to assess real situations, which has enabled teachers to be more empathetic, to process information in such a way that they can perceive, understand and regulate their mood when facing the different events and, thereby, enhance decision-making.

It is clear that emotional intelligence is an area that needs to be worked on and that emotional education programmes need to be developed which meet the key indicators in terms of design, implementation and evaluation (start, follow-up, conclusion). Our having applied a segmentation strategy allows us to state that intervention programmes must be adapted by taking into account the sociodemographic and work profile of the participant group. Moreover, the method considered when designing the programmes must be active, participative, flexible and focused on teachers’ own experience.

The intervention programme based on the Mayer and Salovey model helped to boost teachers’ skills when facing the usual demands of their profession by reducing levels of stress and by having available the fundamental knowledge required to apply emotional education in the classroom. We highlight the importance of continuing to carry out intervention programmes for the different stakeholders involved in the education system, in particular those aimed at teachers. Emotional education is lifelong education which seeks to empower teaching staff through strategies, activities and resources that help them to deal with their thoughts and feelings and to act in the appropriate manner when faced with the different situations they experience.

We believe that it is vital to continue furthering this research topic so as to boost future inquiry that focuses on the design, implementation and evaluation of emotional education programmes that seek to enhance teacher training in order to ensure their general well-being and to help instil emotional education in the classroom and to educate children to be emotionally intelligent. Teachers who participate in the programme can apply emotional education in the classroom and adopt tools, skills, attitudes and behaviours that encourage positive thoughts, explore emotions, offer conflict solving, and that lead to an appropriate personalised teaching and learning process focused on their students. The teachers who took part in the present study expressed greater interest in continuing with their self-training in emotional intelligence, as well as with coping strategies aimed at managing and regulating their emotions and those of their students. Getting teachers to view their emotional education as being key to their overall development and well-being will enable them to take on challenges, make better decisions and to lead a healthier existence with themselves, with their environment and with the people with whom they socialise. Teachers’ emotional intelligence will allow them to improve their learning and their teaching, which will have a direct impact on the overall development of their specialised training.

## Figures and Tables

**Figure 1 ijerph-19-15702-f001:**
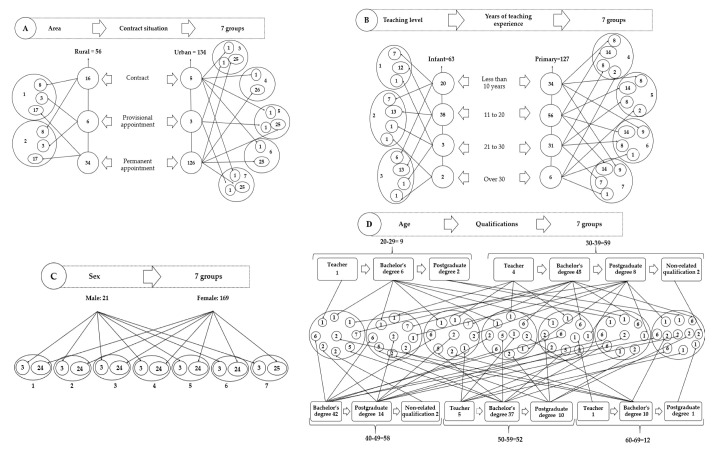
Group segmentation (**A**–**D**) used to carry out the programme. The numbers that were presented in each set of the model correspond to the number of teachers that were assigned to the different segments for the implementation of the program, and thus, the establishment of heterogeneous groups.

**Table 1 ijerph-19-15702-t001:** Session 1. Emotions: How much do I know and identify my emotions?

Emotional Education for Non-College Teachers: Session One			
General Aims of the Session			
To promote a knowledge of emotions amongst teachers from a theoretical and practical perspective. To develop emotional vocabulary. To stimulate the development of emotional perception through emotional regulation strategies to identify and manage emotions in oneself and in others.			
Objectives	Activities	Resources	Orientation Questions
Initial presentation To motivate participating teachers’ active engagement, attendance and permanence.	Presentation of the programme (20 min).	PowerPoint presentation https://drive.google.com/file/d/1V7XcyUVkokxj3KmIDYKvR9A1qKpH5xXJ/view?usp=sharing (Authors’ own elaboration).	What do you know about emotions and about emotional intelligence?
	Dynamics of the presentation “The magic hat of emotions” (40 min).	Random selector https://www.online-stopwatch.com/random-name-pickers/magic-hat-name-picker (accessed on 6 July 2021)	What feelings did you experience with the activity?
Modelling Establishing the difference between the constructs and the importance of perception when developing emotional intelligence.	Teachers’ exposition (30 min).	Presentation in PowerPoint https://drive.google.com/file/d/1g9V2RwJCa3jVnO8MzaKDDD8-W9sLrohb/view?usp=sharing (Authors’ own elaboration)	What is emotion? What is emotional perception?
Guided practice Experiencing and identifying emotions during the relaxation exercises.	Relaxation exercise and introspection (Montoya-Castilla et al., 2021 [4]) (30 min).	Reading https://drive.google.com/file/d/1N2CTA9XySVeHogEHLb8Px2Ujy2N-7HOL/view?usp=sharing (Authors’ own elaboration. Adapted from Montoya-Castilla et al., 2021 [4])).	Are you ready to focus attention on what your body feels when you experience different emotions?
Feedback Identifying, analysing and reflecting on the different feelings the body experiences during relaxation exercises.	Completing the file. Identifying each component that is activated with emotions (30 min).	File https://drive.google.com/file/d/1rRiukwNykC0T77yRZfiPmrbZx-GEi84v/view?usp=sharing (Authors’ own elaboration)	What emotions did you feel during the relaxation and introspection exercise?
Conclusions Establishing the knowledge obtained during the session.	Teamwork (30 min).		What did you learn about the constructs related to emotions and about emotional perception?
Autonomous Work (60 min)			
Objective	Activities	Resources	Orientation Questions
Boosting the knowledge acquired during the first session, through practical exercises that stimulate the development of emotional perception.	Performing deep-breathing exercise 4–7–8, five minutes each day.	Comfortable surroundings	What feelings did you notice with the exercise?
	Taking notes on the emotions, reactions and behaviours you experience in a situation.	Emotional diary (notebook).	Were you able to identify and name the emotions you experienced? Are you happy with your response or behaviour when faced with the emotions you experienced? What could be improved?
	Watch Pilar Fernández’s video “Emotional intelligence in teachers”.	Video https://www.youtube.com/watch?v=6L8iruZGI_8 (accessed on 6 July 2021)	What could you gain from the video to improve your personal life and professional practice?

**Table 2 ijerph-19-15702-t002:** Correlation matrix between the factors that make up emotional intelligence in the experimental group and the control group.

Factors	Experimental Group			Control Group		
	Emotional Perception (Pre)	Emotional Understanding (Pre)	Emotional Regulation (Pre)	Emotional Perception (Pre)	Emotional Understanding (Pre)	Emotional Regulation (Pre)
Emotional perception (post)	0.43	0.18	0.11	0.49	0.14	0.10 *
Emotional understanding (post)	0.17	0.50	0.24	0.16	0.47	0.28
Emotional regulation (post)	0.04 **	0.40	0.27	0.19	0.31	0.32

Note. * *p*-Value < 0.1. ** *p*-Value < 0.05.

**Table 3 ijerph-19-15702-t003:** Influence of the variables that make up the sociodemographic profile in emotional intelligence in the control group and in the experimental group from pretest and post-test data.

Factors		Emotional Intelligence					
		Emotional Perception		Emotional Understanding		Emotional Regulation	
		Pretest	Post-Test	Pretest	Post-Test	Pretest	Post-Test
Gender							
Male	Exp.	29.17 (6.33)	34.67 (3.39)	35.83 (3.81)	36.5 (1.22)	32.67 (9.04)	37.17 (3.19)
	Ctrl.	26.25 (6.65)	23.25 (6.60)	25.75 (4.64)	28.25 (8.10)	32.00 (5.71)	31.75 (5.56)
*p*-Value	Exp.	0.1		0.93		0.42	
	Ctrl.	0.8824		0.77		1.00	
Female	Exp.	29.54 (6.52)	33.36 (3.81)	33.74 (5.03)	35.77 (4.16)	34.66 (4.50)	35.95 (4.04)
	Ctrl.	30.01 (6.33)	29.60 (6.78)	34.35 (4.72)	34.33 (5.07)	35.41 (4.56)	34.50 (4.49)
*p*-Value	Exp.	0.0 **		0.0 **		0.0028 **	
	Ctrl.	0.6295		0.83		0.06	
Age							
Up to 29	Exp.	29.86 (9.28)	33.00 (4.28)	36.86 (3.98)	35.71 (5.25)	36.00 (3.56)	35.28 (5.31)
	Ctrl.	28.29 (6.50)	28.71 (6.90)	34.43 (5.31)	34.29 (5.74)	36.00 (2.83)	34.29 (4.35)
*p*-Value	Exp.	0.6533		0.8957		1.00	
	Ctrl.	0.949		0.847		0.483	
30–39	Exp.	29.51 (6.97)	33.22 (3.76)	34.00 (4.90)	36.30 (3.78)	34.58 (4.27)	35.92 (3.95)
	Ctrl.	30.93 (5.77)	30.08 (6.52)	34.82 (4.83)	34.39 (5.87)	36.47 (3.98)	34.75 (4.92)
*p*-Value	Exp.	0.0030 **		0.0078 **		0.05665	
	Ctrl.	0.5619		0.99		0.059	
40–49	Exp.	29.67 (6.52)	33. 31 (3.62)	32.88 (5.40)	35.03 (4.75)	34.54 (4.86)	35.44 (4.35)
	Ctrl.	28.63 (7.14)	28.67 (7.01)	33.31 (5.06)	34.40 (4.57)	34.37 (4.82)	34.69 (3.96)
*p*-Value	Exp.	0.0012 **		0.0091 **		0.2462	
	Ctrl.	0.922		0.2642		0.9557	
50–59	Exp.	28.9 (5.42)	32.9 (3.74)	33.52 (4.49)	35.62 (4.01)	34.15 (5.26)	36.3 (3.43)
	Ctrl.	30.58 (6.21)	29.73 (7.56)	33.70 (4.76)	33.45 (4.99)	34.82 (4.93)	33.88 (4.60)
*p*-Value	Exp.	0.00055 **		0.02597 *		0.063	
	Ctrl.	0.807		0.8169		0.315	
60 or over	Exp.	31.00 (5.93)	37.40 (3.34)	37.70 (3.13)	38.2 (2.15)	35.90 (4.56)	39.30 (1.49)
	Ctrl.	30.13 (5.03)	29.13 (5.60)	35.62 (4.50)	34.0 (5.40)	34.38 (6.21)	32.63 (4.90)
*p*-Value	Exp.	0.011 *		0.9048		0.04486 *	
	Ctrl.	0.75		0.49		0.4936	
Number of children							
No children	Exp.	27.65 (6.64)	32.56 (3.35)	33.13 (5.92)	36.48 (3.13)	32.69 (4.60)	36.48 (3.30)
	Ctrl.	29.04 (6.91)	28.59 (7.05)	32.09 (5.83)	33.00 (5.53)	34.86 (4.59)	33.54 (4.25)
*p*-Value	Exp.	0.0034 **		0.052		0.0047 **	
	Ctrl.	0.8141		0.62		0.2732	
1 child	Exp.	29.28 (5.69)	33.03 (3.26)	34.33 (5.25)	35.67 (3.79)	34.69 (4.35)	35.89 (4.39)
	Ctrl.	30.01 (5.75)	30.00 (6.42)	34.68 (5.50)	34.42 (4.50)	34.58 (6.56)	34.77 (3.74)
*p*-Value	Exp.	1.00		0.359		0.11	
	Ctrl.	0.0032 **		0.5237		0.30	
2 children	Exp.	30.89 (6.30)	33.57 (4.01)	34.70 (4.60)	36.39 (3.48)	34.68 (4.86)	35.82 (4.14)
	Ctrl.	30.09 (7.20)	29.15 (7.06)	34.41 (4.83)	35.06 (4.59)	35.74 (3.80)	35.37 (4.03)
*p*-Value	Exp.	0.03 *		0.049 *		0.194	
	Ctrl.	0.5		0.5		0.82	
3 children	Exp.	28.68 (6.90)	33.05 (3.96)	32.57 (4.76)	34.81 (4.91)	34.81 (4.86)	35.91 (3.94)
	Ctrl.	29.83 (6.14)	29.67 (6.61)	34.17 (3.87)	33.71 (5.32)	35.67 (3.75)	34.02 (4.94)
*p*-Value	Exp.	0.0009 ***		0.004 ***		0.2589	
	Ctrl.	0.883		0.959		0.1369	
4 children	Exp.	30.57 (6.85)	35.57 (2.56)	35.00 (5.05)	36.07 (4.61)	36.07 (3.79)	35.86 (4.40)
	Ctrl.	31.62 (4.74)	29.85 (8.39)	35.23 (5.29)	34.23 (7.88)	35.15 (5.03)	33.54 (6.39)
*p*-Value	Exp.	0.03354 *		0.5597		0.90	
	Ctrl.	0.8572		0.96		0.7175	
5 or more children	Exp.	35.50 (0.71)	40.00 (1.78)	35.00 (4.24)	36.00 (1.78)	34.78 (2.23)	33.78 (4.01)
	Ctrl.	25.00 (2.17)	26.14 (4.36)	25.12 (4.66)	26.13 (2.74)	25.17 (4.01)	25.99 (2.07)
*p*-Value	Exp.	0.02 *		0.227		0.35	
	Ctrl.	1.00		1.00		0.98	
Professional Qualification							
Teacher	Exp.	29.50 (3.02)	33.67 (3.44)	35.33 (3.44)	36.5 (2.07)	36.33 (1.63)	34.33 (5.04)
	Ctrl.	24.83 (6.04)	26.33 (5.85)	26.83 (4.99)	30. 67 (5.12)	30.17 (5.07)	31.67 (3.26)
*p*-Value	Exp.	0.063		0.76		0.74	
	Ctrl.	0.47		0.26		0.5725	
Bachelor’s degree	Exp.	29.44 (6.63)	33.38 (3.91)	33.75 (5.03)	35.83 (4.10)	34.60 (4.75)	36.18 (3.86)
	Ctrl.	30.58 (6.20)	29.72 (6.76)	34.45 (5.11)	34.59 (4.34)	35.80 (4.20)	34.65 (4.60)
*p*-Value	Exp.	0.00 **		0.0 **		0.002 **	
	Ctrl.	0.3056		0.788		0.67	
Master’s degree	Exp.	30.44 (6.35)	33.84 (3.26)	34.20 (4.61)	36.16 (3.77)	33.92 (4.40)	36.12 (4.16)
	Ctrl.	27.63 (6.50)	29.63 (6.32)	34.00 (6.04)	33.94 (4.87)	34.68 (5.54)	34.10 (3.96)
*p*-Value	Exp.	0.044 *		0.106		0.037 *	
	Ctrl.	0.4382		0.7688		0.36	
Qualifications not related to teaching	Exp.	27.00 (10.14)	29.33 (1.53)	31.33 (10.26)	31.67 (6.66)	34.67 (8.38)	30.66 (6.11)
	Ctrl.	33.50 (2.12)	30.00 (11.31)	32.00 (2.83)	30.50 (12.02)	31.50 (3.53)	32.50 (6.36)
*p*-Value	Exp.	1.0		1.0		0.4	
	Ctrl.	1.0		1.0		1.0	

Note. * *p*-Value < 0.1. ** *p*-Value < 0.05. *** *p*-Value < 0.01.

**Table 4 ijerph-19-15702-t004:** Influence of the variables that make up the work profile in emotional intelligence in the control group and in the experimental group from pretest and post-test data.

Factors		Emotional Intelligence					
		Emotional Perception		Emotional Understanding		Emotional Regulation	
		Pretest	Post-Test	Pretest	Post-Test	Pretest	Post-Test
Contract situation							
Under contract	Exp.	28.47 (6.14)	33.47 (3.4)	35.53 (4.49)	36.79 (2.53)	35.47 (3.85)	37.37 (3.00)
	Ctrl.	31.71 (5.65)	29.59 (8.82)	34.06 (4.84)	32.41 (819)	36.41 (3.67)	33.53 (6.24)
*p*-Value	Exp.	0.012 *		0.58		0.098	
	Ctrl.	0.78		1.0		0.19	
Temporary appointment	Exp.	32.00 (4.27)	35.90 (3.67)	35.50 (3.47)	36.10 (4.48)	35.80 (2.82)	35.30 (5.62)
	Ctrl.	28.75 (4.74)	30.88 (3.68)	34.62 (3.70)	35.13 (3.87)	34.38 (4.27)	33.75 (3.24)
*p*-Value	Exp.	0.044 *		0.6186		0.59	
	Ctrl.	0.53		0.83		0.43	
Definitive appointment	Exp.	29.50 (6.64)	33.24 (3.81)	33.50 (5.11)	35.65 (4.23)	34.42 (4.84)	35.87 (4.00)
	Ctrl.	29.81 (6.50)	29.35 (6.72)	34.11 (4.99)	34.34 (4.80)	35.25 (4.72)	34.58 (4.34)
*p*-Value	Exp.	0.00 **		0.00 **		0.0065 **	
	Ctrl.	0.56		0.75		0.094	
Level taught							
Infant education	Exp.	30.08 (6.07)	33.10 (3.90)	34.28 (4.64)	35.95 (3.84)	34.70 (4.21)	36.03 (3.72)
	Ctrl.	30.57 (6.82)	29.86 (6.68)	34.01 (4.59)	34.23 (4.87)	35.00 (4.16)	34.69 (4.38)
*p*-Value	Exp.	0.0025 **		0.045 *		0.09	
	Ctrl.	0.46		0.67		0.851	
Primary education	Exp.	29.25 (6.62)	33.32 (3.72)	33.98 (5.12)	35.81 (4.18)	34.74 (4.85)	36.15 (3.85)
	Ctrl.	29.70 (5.97)	29.20 (7.45)	34.07 (4.81)	33.63 (5.69)	35.43 (4.95)	34.89 (4.67)
*p*-Value	Exp.	0.0 **		0.019 *		0.055	
	Ctrl.	0.97		0.88		0.08	
Area in which the school where the teacher works is located							
Urban	Exp.	30.70 (3.97)	32.90 (5.45)	33.07 (7.81)	33.23 (5.55)	32.12 (4.95)	32.76 (4.57)
	Ctrl.	29.30 (5.97)	29.60 (3.12)	34.07 (1.12)	33.78 (3.12)	35.12 (4.95)	34.21 (4.12)
*p*-Value	Ctrl.	0.038 *		0.23		0.78	
	Exp.	0.097		0.096		0.23	
Rural	Exp.	29.54 (6.52)	33.36 (3.80)	33.74 (5.03)	35.77 (4.16)	34.66 (4.49)	35.95 (4.04)
	Ctrl.	30.06 (6.33)	29.60 (6.78)	34.35 (4.72)	34.33 (5.07)	35.41 (4.56)	35.49 (4.49)
*p*-Value	Exp.	0.00 **		0.00 **		0.0028 **	
	Ctrl.	0.62		0.82		0.76	
Category of teaching rank (pay grade)							
F-G	Exp.	29.26 (6.55)	33.39 (3.81)	33.67 (5.01)	35.87 (4.00)	34.68 (4.56)	36.06 (4.10)
	Ctrl.	30.39 (6.04)	29.78 (6.41)	34.13 (4.82)	34.09 (5.11)	35.54 (4.36)	35.50 (4.55)
*p*-Value	Exp.	0.00 **		0.00 **		0.0022 **	
	Ctrl.	0.44		0.92		0.061	
D-E	Exp.	30.12 (5.30)	33.41 (3.42)	34.24 (4.44)	34.06 (5.13)	34.59 (4.11)	34.94 (3.63)
	Ctrl.	27.33 (7.93)	27.13 (9.29)	35.13 (5.39)	36.00 (4.86)	34.47 (6.02)	33.80 (4.18)
*p*-Value	Exp.	0.024 *		0.86		0.85	
	Ctrl.	0.90		0.53		0.35	
A-B-C	Exp.	32.90 (7.00)	33.40 (4.43)	35.20 (4.59)	37.30 (3.09)	33.20 (7.19)	36.60 (3.01)
	Ctrl.	27.37 (7.33)	28.00 (8.47)	32.38 (5.42)	32.25 (7.23)	33.25 (5.63)	34.38 (5.09)
*p*-Value	Exp.	0.81		0.37		0.22	
	Ctrl.	0.83		0.91		0.75	
Years of work experience							
<10 years	Exp.	28.98 (6.86)	33.03 (3.48)	34.60 (4.60)	36.27 (3.91)	35.36 (3.68)	36.38 (3.58)
	Ctrl.	31.64 (5.60)	30.42 (6.07)	34.65 (4.78)	34.24 (5.79)	36.04 (3.80)	35.04 (4.46)
*p*-Value	Exp.	0.00059 **		0.048 *		0.09	
	Ctrl.	0.25		0.97		0.27	
11 to 20 years	Exp.	29.62 (6.53)	33.68 (3.92)	33.17 (5.28)	35.32 (4.32)	34.16 (4.90)	35.57 (4.40)
	Ctrl.	29.00 (6.62)	29.05 (7.18)	34.08 (4.77)	34.54 (4.56)	35.48 (4.33)	34.44 (4.36)
*p*-Value	Exp.	0.00 **		0.001 **		0.031 *	
	Ctrl.	0.83		0.52		0.09	
21 to 30 years	Exp.	30.14 (5.49)	32.89 (4.04)	33.61 (4.96)	35.87 (3.83)	33.83 (6.07)	36.08 (3.50)
	Ctrl.	28.50 (6.58)	27.62 (7.57)	32.25 (6.21)	32.38 (6.25)	32.56 (7.37)	33.00 (5.14)
*p*-Value	Exp.	0.14		0.13		0.27	
	Ctrl.	0.80		0.97		0.92	
>30	Exp.	32.00 (6.32)	35.20 (4.02)	37.20 (1.92)	38.6 (1.34)	37.00 (2.45)	39.00 (1.73)
	Ctrl.	32.60 (5.22)	31.40 (5.94)	35.40 (2.70)	33.20 (5.40)	34.20 (3.49)	32.20 (5.36)
*p*-Value	Exp.	0.46		0.23		0.75	
	Ctrl.	0.67		0.67		0.18	
Administrative management							
Yes	Exp.	28.91 (5.87)	33.34 (3.82)	33.61 (4.66)	36.48 (3.99)	34.22 (5.64)	36.17 (3.96)
	Ctrl.	30.84 (6.71)	30.00 (6.39)	33.95 (5.02)	33.47 (5.63)	33.37 (6.88)	33.63 (4.31)
*p*-Value	Exp.	0.0047 **		0.031 *		0.263	
	Ctrl.	0.64		0.81		0.60	
No	Exp.	29.62 (6.59)	33.41 (3.80)	33.83 (5.06)	35.69 (4.12)	34.65 (4.54)	35.96 (4.04)
	Ctrl.	29.85 (6.31)	29.37 (6.90)	34.16 (4.89)	34.27 (5.17)	35.59 (4.17)	34.53 (4.55)
*p*-Value	Exp.	0.0 **		0.0005 **		0.0042 **	
	Ctrl.	0.633		0.67		0.067	

Note. * *p*-Value < 0.1. ** *p*-Value < 0.05.

## Data Availability

The data presented in this study are available from the corresponding authors upon request.

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
