# Peer review of "Group Segmentation as a Strategy for Implementing the Intervention Programme in Emotional Education for Infant and Primary Teachers"

_ijerph, 2022, doi:10.3390/ijerph192315702_

Round 1

Reviewer 1 Report

I consider that the topic of the study is really cutting-edge, being the research problem a very relevant one. The theoretical framework is clearly and correctly defined, supported by well-founded assumptions; I also value positively the methodological part.

However, the educational implications of the study could also be addressed, given their importance in the field of education in today's society and at the present time, being coherent with the analyses carried out and the results obtained. At the same time, it would be possible to include what is new in this study in relation to the work already published on this subject.

Author Response

Consulte el archivo adjunto.

Reviewer 2 Report

What are the study limitations and suggestions for further research? (Authors should add this information at the end of the results discussion). 

Also, please see the attached file.

Reviewer 3 Report

This article is considered interesting, and this study is appropriate due to the importance of development of emotional education both for students and teachers.

This manuscript is well written, and the methodology is suitable for the study in question but the key ideas should be more well presented and discussed.

Indeed, there are some aspects that deserve a second thought to evaluate its appropriateness, and some minor issues to revise. Here are some suggestions to consider:

1-      the objectives must be clear and well defined both in the abstract and in the introduction. I understand that you have 2 goals: test the effectiveness of the program through group segmentation and to determine significant differences. This must be more clear for the understanding of the reader.

2-      Explain what is group segmentation strategy and why will be used for this study. Complement with references of other studies done before.

3-      There is no definition of emotional intelligence. Consider to introduce a simple definition, and than the importance to study it.

4-      The programme is based Mayer and Salovey model. In this article is considered 3 factors, but the model has 4. There is a reason for that?

5-      The results only presente tables. I understand that this were the results obtained. Consider to make some sintesis for each table in order to facilitate the readers understanding.

6-      Discussion point should be divided according to your most important and/or relevant findings. Consider to introduce some sub titles.

7-      In conclusions, line 469,470: “…and when compared to those from previous studies…” consider to introduce some references to evidence this setence.
